# DoraemonGPT⊙ : Toward Understanding Dynamic Scenes with Large Language Models

## ABSTRACT

The field of AI agents is advancing at an unprecedented rate due to the capabilities of large language models (LLMs). However, LLM-driven visual agents mainly focus on solving tasks for the image modality, which limits their ability to understand the dynamic nature of the real world, making it still far from real-life applications, *e.g.*, guiding students in laboratory experiments and identifying their mistakes. Considering the video modality better reflects the ever-changing and perceptually intensive nature of real-world scenarios, we devise DoraemonGPT, a comprehensive and conceptually elegant system driven by LLMs to handle dynamic video tasks. Given a video with a question/task, DoraemonGPT begins by converting the input video with massive content into a symbolic memory that stores *task-related* attributes. This structured representation allows for *spatial-temporal* querying and reasoning by sub-task tools, resulting in concise and relevant intermediate results. Recognizing that LLMs have limited internal knowledge when it comes to specialized domains (*e.g.*, analyzing the scientific principles underlying experiments), we incorporate plug-and-play tools to assess external knowledge and address tasks across different domains. Moreover, we introduce a novel LLM-driven planner based on Monte Carlo Tree Search to efficiently explore the large planning space for scheduling various tools. The planner iteratively finds feasible solutions by backpropagating the result's reward, and multiple solutions can be summarized into an improved final answer. We extensively evaluate DoraemonGPT in dynamic scenes and provide in-the-wild showcases demonstrating its ability to handle more complex questions than previous studies.

## 1 INTRODUCTION

Based on the advancements in large language models (LLMs) [1–8], recent LLM-driven agents [9–11] have demonstrated promise in planning the decomposition of complex image tasks into manageable subtask sequences and solving them step-by-step. While static images have been extensively studied, real-world environments are inherently dynamic [12] and ever-changing [13]. Commonly, capturing real-life scenes is a data-intensive procedure, usually processed by streaming static images into dynamic videos. In turn, the *spatial-temporal* reasoning of videos is critical in real-life recognition, semantic description, causal reasoning, *etc*.

Toward understanding dynamic scenes, developing an LLM-driven agent to handle dynamic videos is of great significance yet involves several grand challenges: **i) Spatial-temporal Reasoning**. The ability to reason about the relationships between instances and actions is crucial for intelligent action planning and decision making. Such relationships may be relevant to space [14], time [15], or their *spatial-temporal* combination. **i) Larger Planning Space**. Compared to static images, high-level semantics about actions and their intentions can typically only be inferred from temporal visual observations [16]. In other words, the inference of temporal semantics is necessary and will enlarge the search space of planning dynamic video tasks by the agent. **iii) Limited Internal Knowledge**. It's clear that LLMs can not encode all the knowledge required for understanding every video due to the ever-changing nature of the real world and/or the lack of training on proprietary datasets [17].

In light of the foregoing discussions, we present DoraemonGPT, an intuitive yet versatile system driven by LLMs that is compatible with various foundation models and video applications. DoraemonGPT has three desirable abilities: **First**, collecting information regarding the given task before

reasoning. In DoraemonGPT, the decomposition of the given dynamic task is decided by the agent-based reasoning of *spatial-temporal* relations, which are inferred from informative attributes, such as instance locations, actions, scene changes, *etc*. However, it is important to note that only task-solving related information is critical, as gathering excessive context tends to hinder the LLMs' capability [18]. **Second**, exploring better solutions before making decisions. LLM-driven planning [19–23] decomposes high-level tasks into sub-tasks or action sequences. Considering an action sequence as a root-to-leaf path in a tree containing all possible sequences, the planning can be viewed as finding optimal decisions from a tree-like search space [24]. Regarding the large planning space for solving tasks in dynamic scenes, prompting LLMs with tree-like search methods [25–27] offers opportunities for better solutions and even the possibility of considering tasks from different perspectives. **Third**, supporting knowledge extension. Just as humans consult reference books to tackle domain-specific issues, DoraemonGPT is designed to select the most relevant knowledge source from a series of given external knowledge sources (*e.g.*, search engines, textbooks, databases, *etc*.) and then query the information from it during the planning.

More specifically, DoraemonGPT is structured in a form of ⟨*memory, tool, planner*⟩ (Fig. 1c): **i)** **Task-related Symbolic Memory (§3.1).** To collect information related to the given video and task, we consider decoupling *spatial-temporal* attributes into two memories: *space-dominant* and *time-dominant*. Before constructing these memories, LLMs are used to determine their relevance to the given task and keep only the useful one(s). Foundation models are then employed to extract *space-dominant* attributes (*e.g.*, instance detection, trajectory, description, action, *etc*.) or *time-dominant* attributes (*e.g.*, frame captions, video descriptions, audio speech, *etc*.) and integrate them into a compact table, which facilitates LLMs to query information by using symbolic language (*e.g.*, SQL language). **ii) Sub-task (§3.1) and Knowledge (§3.2) Tools.** To compact our planner's context/text length and improve effectiveness, we simplify memory information querying by designing a series of sub-task tools. Each tool focuses on different kinds of *spatial-temporal* reasoning (*e.g.*, *"How..."*, *"Why..."*, *etc*.) by using individual LLM-driven sub-agents with task-specific prompts and examples. Additionally, for tasks requiring domain-specific knowledge, external knowledge sources can be easily incorporated through dedicated sub-agent tools. **iii) Monte Carlo Tree Search (MCTS) Planner (§3.3).** To efficiently explore the large planning space, we propose a novel tree-search-like planner. The planner iteratively finds feasible solutions by backpropagating the answer's reward and selecting a highly expandable node to expand a new solution. After summarizing all the results, the planner derives an informative final answer. To design the tree search planner, we equip our DoraemonGPT with MCTS [27–29], which has shown effectiveness in finding optimal decisions from a large search space [30], especially in the game AI community [31–33].

Combining the above designs together, DoraemonGPT handles dynamic *spatial-temporal* tasks effectively and targeted, supports a comprehensive exploration of multiple potential solutions, and can extend its expertise by leveraging multi-source knowledge. On the NExT-QA [34] benchmark, which includes a variety of dynamic scenarios and challenging video reasoning tasks, our DoraemonGPT outperforms recent LLM-driven competitors (*e.g.*, surpassing ViperGPT [9] by **19.3%/5.6%/15.2%** in causal/temporal/descriptive reasoning). Extensive ablation studies validate our MCTS planner's effectiveness, outperforming the naïve DFS method and other baselines. Moreover, when dealing with more complex in-the-wild tasks previously unapplicable or neglected by recent approaches [9, 35], DoraemonGPT provides reasonable answers by integrating external knowledge and summarizing results from multiple feasible solutions.

## 2 RELATED WORK

**Multimodal Understanding.** Before the emergence of LLMs, various efforts were made to create multimodal systems tailored for specific tasks [38–50]. While these systems showed impressive performance in their respective domains, their applicability to broader, real-world scenarios was limited due to the lack of generalizability. Recent years have witnessed remarkable progress in *general* multimodal systems, due to the fast evolution of data volumes and computational resources. Specifically, Frozen [51] is a milestone; it demonstrates a feasible way to empower LLMs the ability to handle visual inputs. Since it was proposed, numerous efforts have been devoted to build large multimodal models [7, 52–54]. Considering the training cost, several attempts [55, 56] try to build zero-shot systems for various tasks. An alternative strategy [57], which will be detailed latter, involves combining multiple models or APIs to tackle compositional multimodal reasoning tasks. Our DoraemonGPT

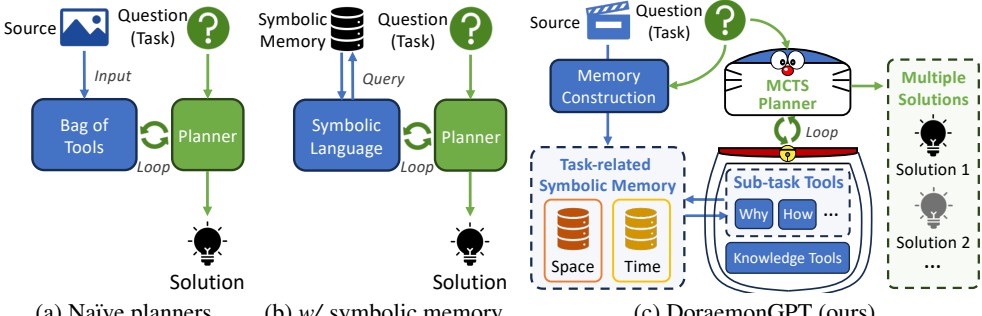

(a) Naïve planners  (b) *w/* symbolic memory  (c) DoraemonGPT (ours)

Figure 1: (a) Naïve LLM-driven planners (*e.g.*, [9–11]) decompose a static image task to a sequence of tool executions, while real-world environments are inherently dynamic. (b) Planners with symbolic memory (*e.g.*, [17, 36, 37]) iteratively generate symbolic languages to retrieve external knowledge or information. (c) Regarding a given dynamic video task, our DoraemonGPT (§3) decouples spatial-temporal attributes into task-related memories. Instead of generating symbolic languages directly, sub-task (symbolic) tools for different kinds of spatial-temporal reasoning (*e.g.*, *"Why..."*, *"How..."*) and other tools (*e.g.*, for retrieving external knowledge) are planned to solve the task. By introducing the MCTS planner, DoraemonGPT can explore the large planning space, find potential solutions, and deliver an improved final answer.

shares a similar spirit of decomposing complex tasks into simpler ones, but it is designed to solve complicated tasks for dynamic modalities in the real-world senarios.

**LLM-driven Modular Systems.** Deconstructing complex tasks and merging the results from multiple intermediate steps is an innate human ability that drives the scientific and industrial communities [58, 59]. Benefiting from the impressive emergent capabilities of LLMs, VisProg [11] pioneers the idea of addressing complex vision tasks through the decomposition of questions into manageable subtasks. Along this line, tremendous progress has been achieved, which can be divided into two categories according to reasoning styles: i) Reasoning with fixed paths [9–11, 60–62]. They transform the given task into a ordered sequence of subtasks, each addressed by a specific module. For example, ViperGPT [9] treats the solving process as a Python program with manually designed APIs. Similarly, HuggingGPT [10] models task dependencies between numerous foundation models. ii) Reasoning with dynamic paths [19, 63–66]. Considering the intermediate results may not meet expectations, a promising avenue is to perform planning and executing concurrently. Such an interactive paradigm [19] provides a flexible and error-tolerant manner compared to those with fixed paths. Additionally, there are many agents focussing on other domains, *e.g.*, planning in the open-world environments [67–69], tool usage [70, 71], reinforcement learning [20, 72]. This work focuses on computer vision only.

Despite impressive, existing LLM-driven modular systems mainly focus on developing specific strategies to solve composition tasks for *static* modalities, ignoring the fundamental gaps between static and dynamic modalities, which is a pivotal aspect towards achieving artificial general intelligence (AGI) [73]. These works, to some extent, can be seen as a subset of our system. Though exceptions exist [9, 35, 66, 74], in general they are scattered, lacking systematic study, *e.g.*, simply treating the video as a sequence of images [9] or building a chatbot based on pre-extracted information of the given video [35, 74]. In sharp contrast, we take the video and task as a whole, resulting in a compact, *task-related* memory. The reasoning paths of our system are dynamic powered by the MCTS planner. In addition to facilitate the answer searching, MCTS planner has the potential to find multiple possible candidates. This is crucial for questions with open-ended answer.

**LLMs with External Memory.** How to effectively design prompt templates, known as prompt engineering, is of importance for accurate LLM responses [75, 76]. One of the areas in the spotlight is memory-augmented LLMs [36, 37, 69, 77–84]. In general, training-free memories can be divided into: i) Textual memory [69, 85]. In this kind of memory, the long contexts LLMs cannot handle (*e.g.*, books) are stored as embeddings, which can be further retrieved by computing similarity. A typical example is the document question answering shown in LangChain[1]. ii) Symbolic memory. It models memory as structured represtations with corresponding symbolic languages, *e.g.*, codes for programming language [84], execution commands for Excel[2], and structured query language (SQL) for databases [36, 37]. Different from techniques [86–93] that directly extend the context window

---

[1]https://docs.langchain.com/docs/use-cases/qa-docs
[2]https://chatexcel.com/

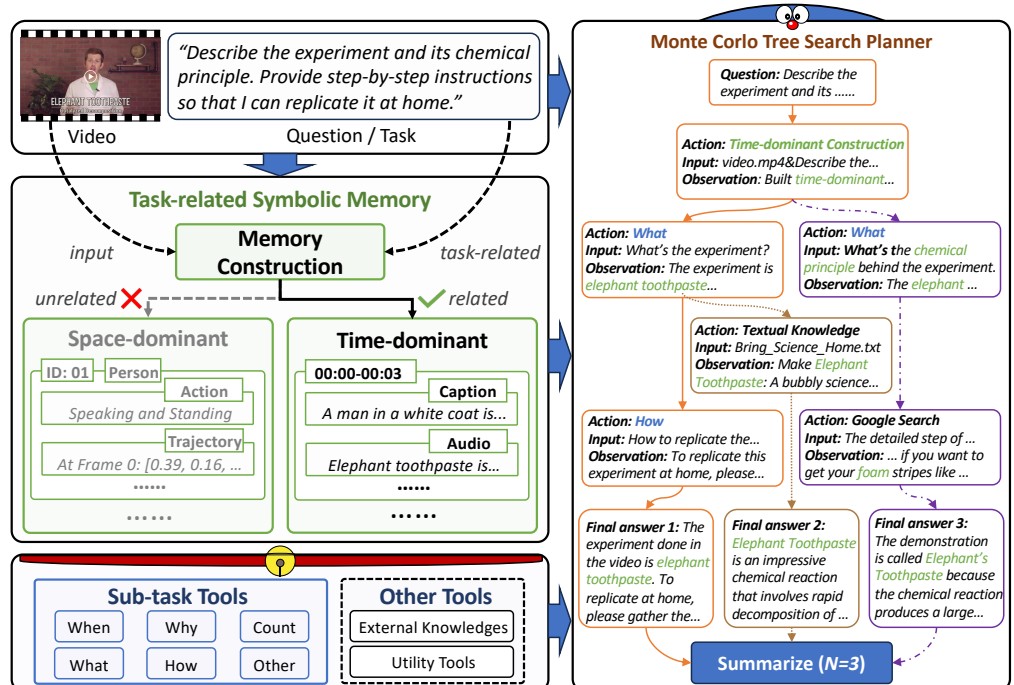

Figure 2: **Overview.** Given a video with a question/task, DoraemonGPT first extracts a Task-related Symbolic Memory (§3.1), which has two types of memory for selection: *space-dominant* memory based on instances and *time-dominant* memory based on time frames/clips. The memory can be queried by sub-task tools, which are driven by LLMs [1] with different prompts and generate symbolic language (*i.e.*, SQL sentences) to do different reasoning. Also, other tools for querying external knowledge (§3.2) or utility tools are supported. For planning, DoraemonGPT employs the MCTS Planner (§3.3) to decompose the question into an action sequence by exploring multiple feasible $N$ solutions, which can be further summarized into an informative answer.

of LLMs, memory-augmented LLMs use retrieval-based approaches to bypass the limitation of context length. This is more favored because: (i) it is a plug-and-play module without any fine-tuning or architectural modifications; and (ii) concurrent works [18, 94] suggest that LLMs may be distracted or lost while encounting irrelevant or long contexts. By absorbing their ideas of memory organization, we construct a request-related database, which stores intance-aware and instance-agnostic information into indivisual tables. To retrieve the relevant information, we explicitly define several sub-task tools based on prompt templates and SQL, with respect to different purposes. With a broader view, our multi-source knowledge, which is a complementary module to provide reliable guidances in specific domains, can also be considered as a hybrid of external memories.

## 3 DORAEMONGPT

**Overview**. DoraemonGPT is an LLM-driven agent capable of seamlessly utilizing various tools to decompose a complex dynamic video task into sub-tasks and solve them. Given a video ($V$) with a textual task/question ($Q$), DoraemonGPT first extracts a Task-related Symbolic Memory (§3.1) from $V$ based on the task analysis of $Q$. Next, employing a Monte Carlo Tree Search (MCTS) Planner (§3.3), DoraemonGPT automatically schedules the tool sets for querying the symbolic memory, accessing external knowledge (§3.2), and calling other utility tools (such as video inpainting, *etc.*) to solve the question $Q$. Ultimately, the planner explores the planning space, returns multiple possible answers, and summarizes an improved answer. An illustration is shown in Fig. 2.

### 3.1 TASK-RELATED SYMBOLIC MEMORY (TSM)

Videos are complex dynamic data, including *spatial-temporal* relations. When giving a question $Q$ regarding a video $V$, only a part of related attributes is essential for making a solution, disregarding the abundance of extraneous information. Thus, we propose to extract and store potentially relevant video information regarding $Q$ into a TSM before solving $Q$.

Table 1: The attribute types in *space-dominant* and *time-dominant* memories (§3.1). Each attribute is extracted or predicted based on different foundation models.

| Attribute | Used Model | Explanation |
|---|---|---|
| *Space-dominant Memory* | | |
| ID number | - | A unique ID assigned to an instance |
| Category | YOLOv8 [95] | The category of an instance, *e.g.*, person |
| Trajectory | Deep OC-Sort [96] | An instance's bounding box in each frame |
| Segmentation | YOLOv8-Seg [95] | An instance's segmentation mask in each frame |
| Appearance | BLIP [54] / BLIP-2 [55] | A description of an instance's appearance |
| Action | InternVideo [97] | The action of an instance |
| *Time-dominant Memory* | | |
| Timestamp | - | The timestap of a frame/clip |
| Audio content | Whisper [98] | Speech recognition results of the video |
| Optical content | OCR [99] | Optical character recognition results of the video |
| Captioning | BLIP [54]/BLIP-2 [55]/ InstructBlip [100] | Frame-level/clip-level captioning results |

**TSM Construction.** To construct the TSM, we use a straightforward in-context learning method [51] to select the task type of TSM based on the question $Q$. We place the `task description` of each type of TSK into the context of our LLM-driven planner, which will be prompted to predict a suitable TSM in the format like *"Action: ⟨TSM_type⟩ construction..."*. Then, the API of constructing the corresponding TSM will be called to extract task-related attributes and store them in an SQL table, which can be accessed by symbolic languages, *i.e.*, SQL sentences.

There is no standardized criterion for categorizing video tasks. In DeoraemonGPT, we choose the perspective of *spatial-temporal* decoupling, which has been widely applied in video representation learning [101–103], to design two memory types:

- *Space-dominant* memory is primarily used to address questions related to specific targets (*e.g.*, persons or animals) or their spatial relations. We use multi-object tracking methods [96] to detect and track instances. Each instance has attributes that include unique `ID`, semantic `category`, `trajectory` & `segmetnation` for localization, `appearance` description extracted by [54, 55] and used for text-based grounding, and `action` classification.
- *Time-dominant* memory focuses on constructing temporal-related information of the video regarding the question. It requires comprehending the content throughout the video. The attributes stored in this memory include `timestamp`, `audio content` by ASR [98], `optical content` by OCR [99], frame-level `captioning` by [54, 55, 100], clip-level `captioning` by deduplicating similar and continuous frame-level results, etc.

Table 1 provides the attribute types with corresponding extraction models of our TSMs.

**Sub-task Tools**. Although LLM-driven agents [35, 36] can assess external information by in-context learning of the whole memory or generating symbolic sentences to access the memory, these methods can significantly increase the length of the context, potentially leading to the omission of crucial information in the reasoning process or being influenced by redundant context. Thus, we provide a series of sub-task tools responsible for querying information from our TSMs [18, 94] by answering sub-task questions. The LLM-driven planner learns the function of each sub-task tool through its in-context `description`, which describes the `sub-task description`, `tool name`, and `tool inputs`. To call the API of a sub-task tool, DoraemonGPT parses the command generated by LLMs, like *"Action: ⟨tool_name⟩. Input: ⟨video_name⟩#⟨sub_question⟩..."*.

To collaborate with the above two kinds of TSMs, we design sub-task tools with different `sub-task description` and for solving different sub-questions, including:

- *When*: related to temporal understanding, *e.g.*, *"When the dog walks past by the sofa?"*
- *Why*: related to causal reasoning, *e.g.*, *"Why did the lady shake the toy?"*
- *What*: describing the required information, *e.g.*, *"What's the name of the experiment?"*
- *How*: what manner, means, or quality of something, *e.g.*, *"How does the baby keep himself safe?"*
- *Count*: counting something, *e.g.*, *"How many people in the room?"*
- *Other*: questions not in the above tools, *e.g.*, *"Who slides farther at the end?"*

The `API functions` of these tools are built upon LLMs as well. Each sub-task tool function is an individual LLM-driven agent, which can generate SQL sentences to query our TSMs and answer

the given sub-task question. Different sub-task agents have different in-context examples regarding their purposes. Note that a sub-question may be suitable for two or more sub-tools (*e.g.*, *"What was the baby doing before playing the toy?"*, related to *what* and *when*), and our MCTS planner (§3.3) is capable of exploring different selections.

## 3.2 KNOWLEDGE TOOLS AND OTHERS

When tackling complex problems, LLM-driven agents sometimes fail to make accurate decisions solely based on video understanding and the implicit knowledge learned by LLMs during training. Thus, DoraemonGPT supports the integration of external knowledge sources that can assist the LLM in comprehending the specialized content within the input video/question. In DoraemonGPT, a knowledge source can be integrated in a plug-and-play manner by using an individual knowledge tool. Similar to the sub-task tools (§3.1), a knowledge tool consists of two parts: **i**) an in-context `knowledge description` to describe the given knowledge source and **ii**) an `API function` to query information from the source by question answering.

We consider three types of `API function` for covering different knowledge: **i**) *symbolic knowledge* refers to information presented in a structured format such as Excel or SQL tables. The `API function` is a symbolic question-answering sub-agent like our sub-task tools (§3.1). **ii**) *textual knowledge* encompasses knowledge expressed through natural language text, such as research publications, reference books, *etc*. The `API function` is built based on text embedding and searching [104]. **iii**) *web knowledge* denotes knowledge searched from the internet. The `API function` is a search engine API, such as Google, Bing, *etc*. Besides the knowledge tools, DoraemonGPT also supports integrating general *utility tools*, commonly used in recent LLM-driven agents [57], to help complete more specialized vision tasks, such as video editing, inpainting, *etc*.

## 3.3 MONTE CARLO TREE SEARCH (MCTS) PLANNER

Previous LLM-driven planners [9–11] decompose the given $Q$ into an action/sub-task sequence and solve it step by step. Such a strategy can be seen as a greedy search method which generates a chain of action nodes until the final answer. However, we consider the large planning space of solving dynamic video tasks as a tree, and a single attempt may not yield the correct result, or better solutions may exist. To efficiently explore the planning space, we propose a novel tree-search-like planner equipped with MCTS [27–29], which has shown its practicality in searching large trees.

We define the question input $Q$ as the root node $v_0$, and an action or tool call is a non-root node, then an action sequence can be viewed as a path from the root node to a leaf node. In our MCTS planner, a non-root node is a ReAct [19]-style step in the form of ⟨*thought, action, action input, observation*⟩, and a leaf node has a *final answer* in addition. The planner iteratively executes the following four phases for $N$ times and produces $N$ solutions:

**Node Selection.** Each iteration starts by selecting an expandable node for planning a new solution. For the first iteration, only the root $v_0$ is selectable. For subsequent iterations, we randomly select a non-leaf node based on their sampling probability, formulated as $P(v_i) = Softmax(R_i)$, where $R_i$ is the reward value of node $v_i$ initialized as 0 and updated in the Reward Back-propagation phase. The node with a higher reward has a greater probability of being selected.

**Branch Expansion.** A child will be added to the selected expandable node to create a new branch. To leverage LLM for generating a new tool call different from the previous child nodes, we add historical tool actions into the prompt of LLM and instruct it to make a different choice. Such an in-context prompt will be removed in the subsequent chain execution toward a new final answer.

**Chain Execution.** After expanding a new branch, we use a step-wise LLM-driven planner [19] to generate a new solution. The execution process consists of a chain of steps/nodes of tool calls and will terminate upon obtaining the final answer or encountering an execution error.

**Reward Back-propagation.** After obtaining a leaf/outcome node $v_l$, we will gradually propagate its reward to its ancestor nodes until $v_0$. In our method, we consider two kinds of reward:

- *Failure*: the planner produces an unexpected result, such as a failed tool call, incorrect result format, *etc*. The reward $R_{v_l}$ is set to a negative value (*e.g.*, $-1$) for these cases.

Table 2: Comparison of our DoraemonGPT with SOTAs on NExT-QA [34] (§4.2). †: reimplement using the officially released codes. *: we filter out those failed executions (*i.e.*, compilation error) of ViperGPT [9] and record the performance on successful executions (840/900 on `s_val`). All LLM-driven systems (LLM agent) use the same LLM, *i.e.*, GPT-3.5-turbo.

| | Method | Split | Publication | | $Acc_C$ | $Acc_T$ | $Acc_D$ | Avg | $Acc_A$ |
|---|---|---|---|---|---|---|---|---|---|
| Supervised | HME [105] | val | CVPR | 2019 | 46.2 | 48.2 | 58.3 | 50.9 | 48.7 |
| | VQA-T [106] | val | ICCV | 2021 | 41.7 | 44.1 | 60.0 | 48.6 | 45.3 |
| | ATP [107] | val | CVPR | 2022 | 53.1 | 50.2 | 66.8 | 56.7 | 54.3 |
| | VGT [108] | val | ECCV | 2022 | 52.3 | 55.1 | 64.1 | 57.2 | 55.0 |
| | VGT [108] | s_val | ECCV | 2022 | 49.7 | 53.3 | 63.7 | 55.6 | 55.6 |
| | MIST [109] | val | CVPR | 2023 | 54.6 | 56.6 | 66.9 | 59.3 | 57.2 |
| | MIST [109] | s_val | CVPR | 2023 | 51.7 | 55.3 | 67.0 | 58.0 | 58.0 |
| LLM agent | †ViperGPT [9] | s_val | ICCV | 2023 | 29.7 | 37.3 | 47.3 | 38.1 | 38.1 |
| | *†ViperGPT [9] | *s_val | ICCV | 2023 | 33.0 | 40.1 | 48.8 | 40.8 | 40.8 |
| | †VideoChat [35] | s_val | - | | 46.7 | 45.3 | 61.0 | 51.0 | 51.0 |
| | Ours | s_val | - | | **52.3** | **45.7** | **64.0** | **54.0** | 54.0 |

- *Non-failure*: the planner successfully produces a result that does not belong to failure results, but it is not sure whether the result is correct as ground truth. $R_{v_l}$ is set to a positive value (*e.g.*, 1).

For simplicity, let $\alpha$ be a positive base reward, we set $R_{v_l} = \pm\alpha$ for *Failure* and *Non-failure*, respectively. According to [94], the outcome produced by LLMs is more related to the context at the beginning (the initial prompts) and the end (the final nodes). We consider that more reward should be applied to nodes close to $v_l$. Thus, the backpropagation function is formulated as $R_{v_i} \leftarrow R_{v_i} + R_{v_l} e^{\beta(1-d(v_i,v_l))}$, where $d(v_i, v_l)$ denotes the node distance between $v_i$ and $v_l$, and $\beta$ is a hyper-parameter for controlling the decay rate of the reward. The further the node distance, the greater the reward decay ratio, $e^{\beta(1-d(v_i,v_l))}$.

After all the MCTS iterations, the planner will produce $N$ *non-failure* answers at most, and we can use LLMs to summarize all the answers to generate an informative answer. Alternatively, for single-/multiple-choice questions, we can determine the final answer through a voting process.

## 4 EXPERIMENT

### 4.1 EXPERIMENTAL SETUP

**Datasets.** NExT-QA [34] is a benchmark for video question answering designed to improve video understanding by focusing on causal action reasoning, temporal action reasoning, and common scene comprehension. It contains 34,132/4,996/8,564 questions for `train`/`val`/`test`. Each question is annotated with a question type (causal/temporal/descriptive) and 5 answer candidates. Limited by the frequency of API calls with GPT-3.5 and budget, we create a balanced subset of NExT-QA as in [11]. Concretely, for evaluation, we randomly sample up to 300 samples per question type from the `val` set, resulting in a total of 900 questions (`s_val`). For ablation studies, we randomly sample 10 samples per question type from `train`, resulting in a total of 30 questions (`s_train`).

**Evaluation Metric.** We adopt the standard metric [34, 109], top-1 test accuracy, for evaluation. We report accuracy for causal ($Acc_C$), temporal ($Acc_T$), descriptive ($Acc_D$). We also report Avg (the average of $Acc_C$, $Acc_T$, and $Acc_D$) and $Acc_A$ (the overall accuracy of all questions).

**Implementation Details.** We use GPT-3.5-turbo API provided by OpenAI as our LLM. As summarized in Table 1, we use BLIP series [55, 100] for captioning, YOLOv8 [95] and Deep OC-Sort [96] for object tracking, PaddleOCR [99] for OCR, InternVideo [97] for action recognition, and Whisper [98] for ASR. Our experiments are conducted under the in-context learning (ICL) setting.

**Competitors.** We involve several *open-sourced* LLM-driven agents for performance comparison. ViperGPT [9] leverages code generation models to create subroutines from vision-and-language models through a provided API, thus it solves the given task by generating Python code that is later executed. VideoChat [35] is an end-to-end chat-centric video understanding system that integrates several foundation models and LLMs to build a chatbot. We do not report the performance of others as they do not release their codes for video tasks or even open-source it.

**Hyperparameters.** We set $\alpha = 1$ and $\beta = 0.5$ for the reward back-propagation (*cf.* §3.3). For experiments on NExT-QA, exploring $N = 2$ solutions has better accuracy-cost trade-offs.

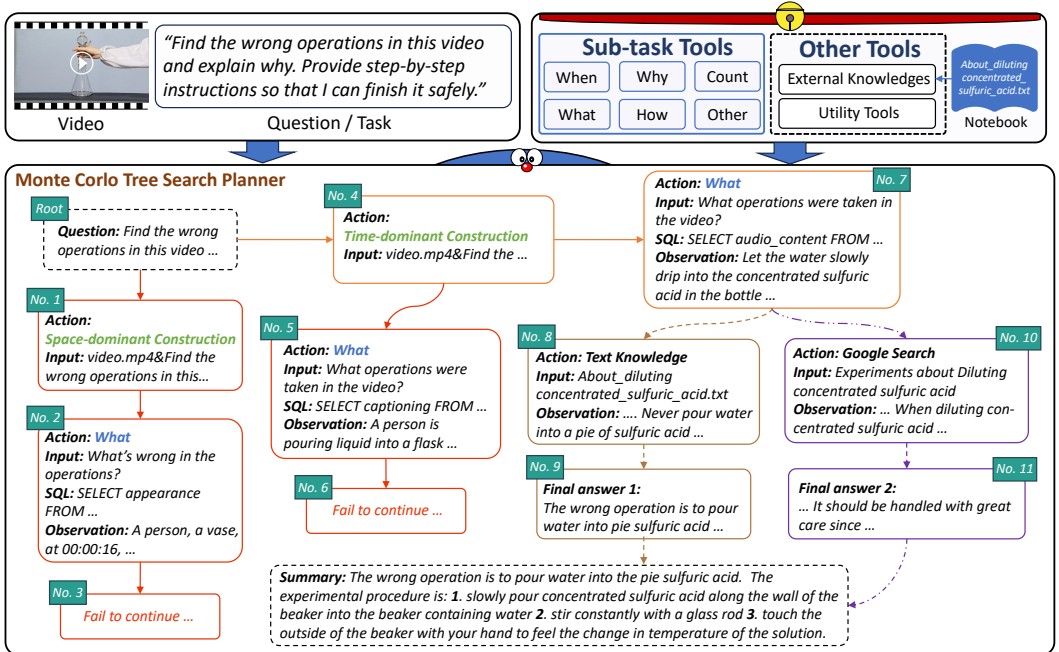

Figure 3: An in-the-wild example of DoraemonGPT. Given a video input and a question, our system automatically explores the solution space powered by MCTS planner and various tools. This figure demonstrates both the utilized tools, and the result of intermediate steps during the exploration. Taking advantage of the tree-like exploration paths, DoraemonGPT can not only summarize collected answers into a better one, but also has the potential to generate multiple potential solutions. (§4.3)

**Reproducibility.** Our algorithm is implemented in PyTorch and LangChain and tested on an NVI-DIA Tesla A100 GPU with 80G memory. To guarantee reproducibility, the code will be released.

## 4.2 QUANTITATIVE RESULT

Table 2 presents comparisons of our DoraemonGPT against several top-leading supervised VQA models and LLM-driven systems on NExT-QA. As can be seen, DoraemonGPT achieves competitive performance compared to recently proposed supervised models. In particular, it shows a more promising improvement in causal questions, even outperforming the previous SOTA, MIST [109] (**52.3** *vs* 51.7). The main reason is that our task-related symbolic memory can provide enough information for reasoning. In terms of temporal and descriptive questions, supervised models are slightly superior to ours, mainly due to their well-designed architectures that have learned underlying patterns. In addition, DoraemonGPT outperforms recent concurrent works, *i.e.*, ViperGPT [9] and VideoChat [35]. Concretely, it outperforms ViperGPT by **19.3/5.6/15.2/13.2** ($Acc_C$/$Acc_T$/$Acc_D$/Avg) and VideoChat by **5.6/0.4/3.0/3.0** across the four question types. Note that we have filtered out the failed execution cases (60/900 on `s_val`) of ViperGPT and only record the performance on successful executions, which improves ViperGPT's Avg score from 38.1 to 40.8. These results demonstrate the efficacy of our MCTS planner based on task-related memory.

## 4.3 IN-THE-WILD EXAMPLE

In Fig. 3, we visualize the reasoning paths of an in-the-wild example. As depicted, DoraemonGPT is asked to check the experimental operations shown in the video and tell the user how to finish it step by step. Specifically, DoraemonGPT first makes two failed attempts at the beginning, *i.e.*, the queried SQL table or symbolic memory does not contain the information related to the sub-question. Regarding the relevant parts, DoraemonGPT understands the experimental operations shown in the video after expanding a new tree branch by querying a different sub-question. Then, there are two alternative ways to get the final answer, *i.e.*, through the textual notebook provided by users or through a search engine. These paths generate two relevant but different final answers, which can be further summarized into a better, more comprehensive answer. Such an exploration process shows our system's ability to handle questions more complex than those constructed in previous studies.

Table 3: A set of ablative experiments about the MCTS planner on NExT-QA [34] s_train (§4.4).

| $N$ | $Acc_C$ | $Acc_T$ | $Acc_D$ | $Acc_A$ |
|---|---|---|---|---|
| 1 | 63.3 | 20.0 | 46.7 | 43.3 |
| 2 | 80.0 | 43.3 | 46.7 | 56.7 |
| 3 | 86.7 | 43.3 | 53.3 | 61.1 |
| 4 | 96.7 | 46.7 | 53.3 | 65.7 |
| 5 | 86.7 | 43.3 | 50.0 | 60.0 |

| $\alpha$ | $\beta$ | $Acc_C$ | $Acc_T$ | $Acc_D$ | $Acc_A$ |
|---|---|---|---|---|---|
| 0.5 | 1.0 | 86.7 | 23.3 | 50.0 | 53.3 |
| 1.0 | 0.5 | 96.7 | 46.7 | 53.3 | 65.7 |
| 0.5 | 2.0 | 86.7 | 26.7 | 50.0 | 54.4 |
| 2.0 | 0.5 | 83.3 | 46.7 | 50.0 | 60.0 |
| 2.0 | 2 | 80.0 | 46.7 | 50.0 | 58.9 |

| Strategy | $Acc_C$ | $Acc_T$ | $Acc_D$ | $Acc_A$ |
|---|---|---|---|---|
| *DFS* | 66.7 | 36.7 | 50.0 | 51.1 |
| *Root* | 73.3 | 16.7 | 46.7 | 45.6 |
| *Uniform* | 67.7 | 26.7 | 50.0 | 47.8 |
| *MCTS* | 96.7 | 46.7 | 53.3 | 65.7 |

(a) Number of answer candidates    (b) Reward and decay rate ($N = 4$)    (c) Exploring strategy ($N = 4$)

## 4.4 DIAGNOSTIC EXPERIMENT

To gain more insights into DoraemonGPT, we conduct a set of ablative experiments on NExT-QA [34] s_train by randomly select 90 videos.

**Task-related Symbolic Memory.** First, we investigate the essential components in DoraemonGPT, *i.e.*, symbolic memories (§3.1) for *space-dominant* (SDM) and *time-dominant* (TDM) information. The results are summarized in Table 4. Two crucial conclusions can be drawn. **First**, TDM is more pre-

Table 4: Analysis of essential components on NExT-QA [34] s_train (§4.4).

| TDM | SDM | $Acc_C$ | $Acc_T$ | $Acc_D$ | $Acc_A$ |
|---|---|---|---|---|---|
| ✓ | | 63.3 | 26.7 | 53.3 | 47.8 |
| | ✓ | 53.3 | 23.3 | 46.7 | 41.1 |
| ✓ | ✓ | 96.7 | 46.7 | 53.3 | 65.7 |

ferred for temporal questions, while SDM can provide relevant information for descriptive questions. **Second**, our complete system achieves the best performance by combining our SDM and TDM, confirming the necessity of dynamically querying two types of symbolic memory.

**Multiple Solutions by MCTS Planner.** We next study the influence of the number of answer candidates during the exploring process of our MCTS planner. When $N = 1$, the planner is equivalent to a greedy search, explores only a chain of nodes, and returns a single answer – the first node in LLM's thinking that can be terminated without further exploration. As shown in Table 3a, gradually increasing $N$ from 1 to 4 leads to better performance (i.e., $43.3 \rightarrow 65.7$). This supports our hypothesis that one single answer is far from enough to handle the larger planning space for dynamic modalities and proves the efficacy of our MCTS planner. Since the questions in NExT-QA [34] are single-choice, exploring more answers does not always result in positive returns. We stop using $N > 5$ as the required number of API calls exceeds our budget.

**Back-propagation in MCTS Planner.** Then, we ablate the effect of the base reward $\alpha$ and decay rate $\beta$ (*cf.* §3.3) that control the exploring procedure of our MCTS planner. As reported in Table 3b, their performance is stable regardless of the combination of $\alpha$ and $\beta$ used. Hence, we set the slightly better one, $\alpha = 1$ and $\beta = 0.5$, as our default setting. We leave some special combinations in the next part (*e.g.*, our MCTS planner becomes the depth-first search (*DFS*) when setting $\beta = 10^8$ and $R_{v_l} = 1$ for both *Failure* and *Non-failure* cases).

**Exploring Strategies used by Planner.** Last, to verify the advantage of our MCTS planner, we compare MCTS with several standard exploring strategies, *i.e.*, depth-first search (*DFS*), *Root*, which always selects the root node, and *Uniform*, which samples a node with equal probability. As shown in Table 3c, we observe that their performance is suboptimal due to the inability to leverage the value/reward of the outcome leaf nodes and accordingly adjust their searching strategy. Compared to these naïve strategies, our MCTS planner adaptively samples a node with the guidance of reward back-propagation, which is more effective in a large solution space. These results further validate the superiority of the proposed MCTS planner.

## 5 CONCLUSION

Regarding real-world scenarios' dynamic and ever-changing nature, we present DoraemonGPT, an LLM-driven agent for solving dynamic video tasks. Compared to existing LLM-driven visual modular systems, DoraemonGPT has merits in: **i**) conceptually elegant system designed by delving into the dynamic modalities in our lives; **ii**) compact task-related symbolic memory by decoupling, extracting, and storing *spatial-temporal* attributes; **iii**) effective and decomposed memory querying through symbolic sub-task tools; **iv** plug-and-play knowledge tools for accessing domain-specific knowledge; **v**) automated exploration of large planning space using MCTS planner, providing multiple solutions with an informative final answer; and **vi**) answer diversity that provides multiple potential candidates by fully exploring the solution space. Experiments confirm the versatility and effectiveness of DoraemonGPT for solving complicated tasks in dynamic scenes.

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

## A APPENDIX

This appendix contains additional details for the ICLR 2024 submission, titled *"DoraemonGPT: Toward Solving Real-world Tasks with Large Language Models"*. The appendix is organized as follows:

- §A.1 depicts visual examples regarding the MCTS planner.
- §A.2 offers more implementation details of the MCTS planner.
- §A.3 introduces more in-the-wild examples.
- §A.4 provides inference results on NExT-QA [34] dataset.
- §A.6 analyzes time of inference and efficiency of token usage.
- §A.5 discusses used foundation models.
- §A.8 discusses our limitations.
- §A.9 discusses the broader impacts of our work.

### A.1 ILLUSTRATION OF MCTS PLANNER

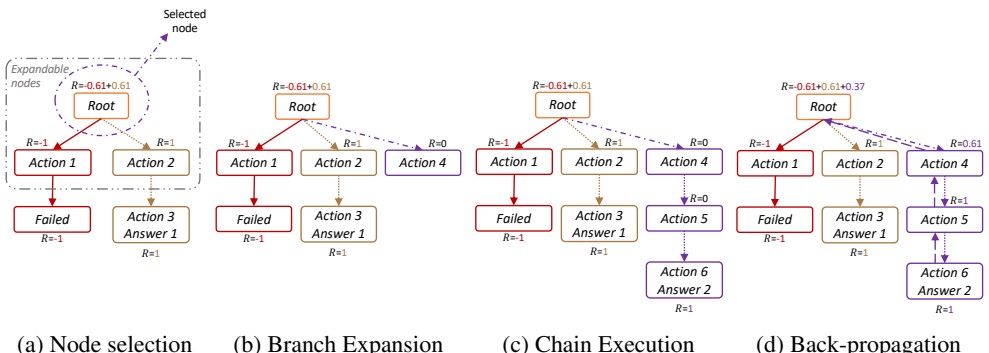

    (a) Node selection     (b) Branch Expansion     (c) Chain Execution     (d) Back-propagation

Figure 4: An illustration of our Monte Carlo Tree Search (MCTS) planner (§A.1). $R$: the reward of a node. *Root*: the input video and question/task. *Action*: a ReAct [19]-style step in the form of ⟨*thought, action, action input, observation*⟩.

Fig. 4 illustrates the MCTS planner with one failed solution and two feasible solutions. The illustrated iteration, which produces the second feasible answer, begins with a *node selection* (Fig. 4a), and the *Root* node with the second highest reward is luckily sampled from all expandable non-leaf nodes. Then, the MCTS planner expands the *Root* node with a new child node, *Action 4*, in *Branch Expansion* (Fig. 4b). Following the expansion, the planner continuously executes actions after *Action 4* until getting a new answer, *Answer 2* (Fig. 4c). Lastly, the planner back-propagates the reward of *Answer 2* to its ancestors. Note that those nodes closer to *Answer 2* receive more rewards.

### A.2 IMPLEMENTATION DETAILS OF MCTS PLANNER

Fig. 5 shows the in-context prompt used in the LLMs of our MCTS planner. By changing the placeholders in the form like {*placeholder*}, the prompt can be adapted to complete **branch expansion** or *chain execution*. The meaning of each placeholder in the prompt is listed below:

- {*video_filename*}: the file path of the input video.
- {*input_question*}: the given question/task regarding the given video.
- {*tool_names*}: the names of tools that can be called by the planner, including sub-task tools, knowledge tools, and utility tools.
- {*tool_descriptions*}: the descriptions of all the callable tools' functions and input format. For example, the description of our *What* sub-task tool is *"Useful when you need to describe the content of a video......The input to this tool must be a string for the video path and a string for the question. For example: inputs is ./videos/xxx.mp4#What's in the video?"*.
- {*agent_scratchpad*}: the place to put the intermediary output during executing a ReAct [19] step.

```
"""
Regarding a given video from {video_filename}, answer the following
    questions as best you can. You have access to the following tools:
{tool_descriptions}
Use the following format:
Question: the input question you must answer
Thought: you should always think about what to do
Action: the action to take, should be one of [{tool_names}]
Action Input: the input to the action
Observation: the result of the action
... (this Thought/Action/Action Input/Observation can repeat N times)
Thought: I now know the final answer
Final Answer: the final answer to the original input question
Begin!
Question: {input_question}
{ancestor_history}
Thought: {expansion_prompt} {agent_scratchpad}
"""
```

Figure 5: The in-context prompt of the MCTS planner (§A.2).

- {*ancestor_history*}: the place to put the history of all the ancestor nodes. For example, when selecting a non-root node for *branch expansion*, the action history (which is a string in the form of ⟨*thought, action, action input, observation*⟩ for each node) of all the ancestor nodes of this non-root node will be put in {*ancestor_history*}.

- {*expansion_prompt*}: the place to put the history of all the child nodes for expanding a node, *e.g.*, *"I have thought about the next action before, such as......I want to think out a different action."*. Only useful in the *branch expansion* phase, set to an empty string for *chain execution*.

### A.3 MORE IN-THE-WILD EXAMPLES

In Fig. 6a, we visualize the reasoning path of a standard video understanding task. As depicted, DoraemonGPT is asked to identify the speaker and analyze information about the dismissal. After several calls to various tools, DoraemonGPT got the right answers. Here we also visualize the *time-dominant* symbolic memory, which is the pivotal part of data processing in DoraemonGPT. Combining it with the well-defined symbolic language (SQL) promises transparency and efficiency.

In addition, we demonstrate an example of video editing by integrating a video inpainting tool. In Fig. 6b, DoraemonGPT is asked to recognize the right person and remove it from the video. To accomplish this, DoraemonGPT constructs the *space-dominant* memory that encompasses the segmentation results for each object within the scene. After recognizing the right person, the inpainting tool is successfully called with an input of the unique ID number assigned to the man on the right, which successfully generates the desired video output.

### A.4 INFERENCE RESULTS ON NEXT-QA

Fig. 7 depicts inference results of DoraemonGPT on NExT-QA [34] dataset. From the top part, we have the following findings: (i) A simple question can be finished within a sub-task tool, *e.g.*, using only the What tool can get the correct answer. (ii) The output of LLM that is not formatted may result in an error case, which is very common in current LLM-driven agents. Similar examples can be observed in the bottom part of the same figure.

As shown in the bottom part of Fig. 7, it's quite possible to pick the wrong tool in the early stages of exploration. Our system is able to explore the planning space with multiple branches further. Interestingly, LLM sometimes considers current information insufficient to make a choice. This is tolerated as our system will eventually vote or summarize all candidate answers.

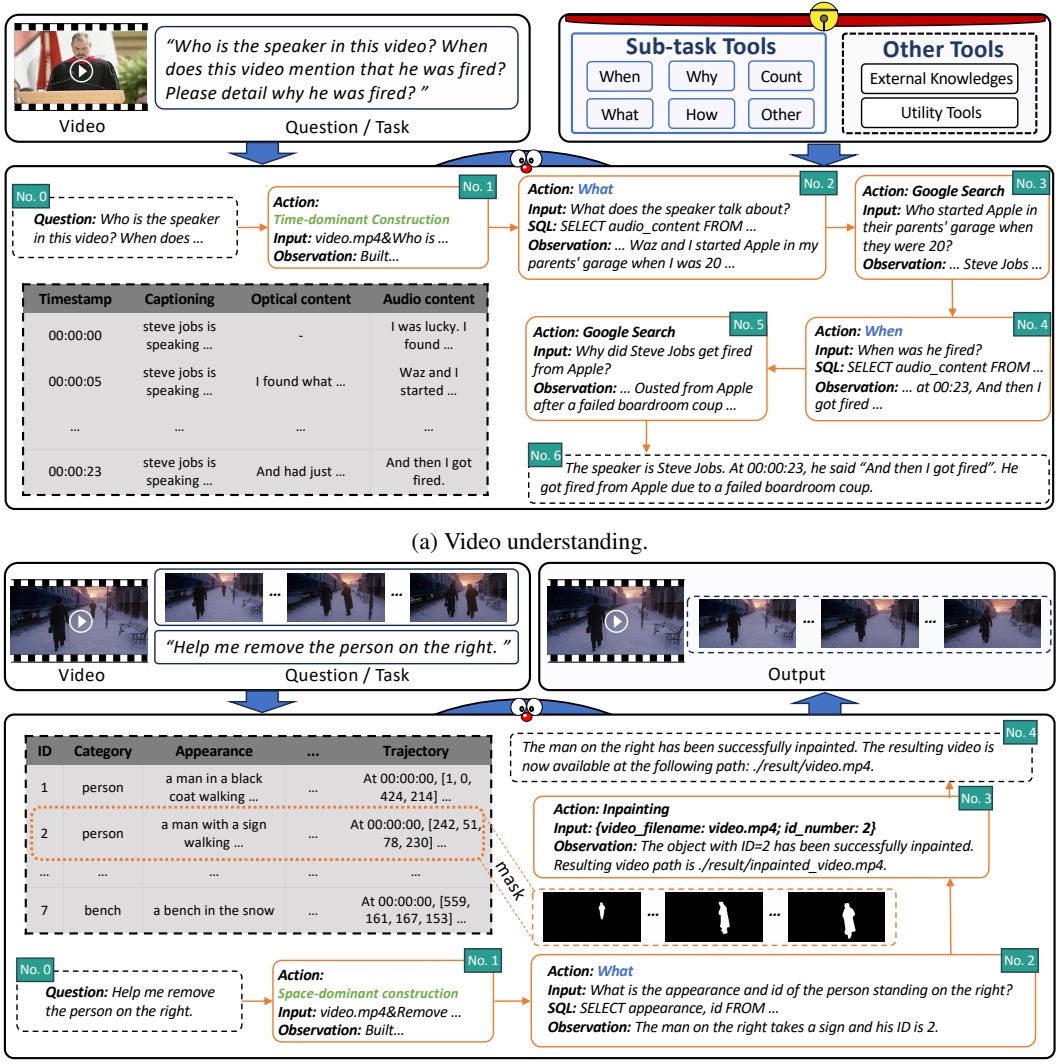

(a) Video understanding.

(b) Video editing.

Figure 6: In-the-wild examples of DoraemonGPT (§A.3). In the video editing example, the segmentation mask is also visualized.

## A.5 DISCUSSION ON THE IMPACT OF FOUNDATION MODELS

DoraemonGPT leverages foundation models to extract space-dominant and time-dominant information from videos. Hence, the performance of DoraemonGPT is influenced by the quality of these models as well as its own limitations. This impact can be further summarized as follows:

**In space-dominant memory:**

**Detection (YOLOv8 [95]):** The object categories (COCO [110], 80 common categories) are limited by the model, which hinders DoraemonGPT from obtaining information about objects outside these categories. However, YOLOv8 [95] can be replaced with a detection model that supports a wider range of categories (such as one trained on LVIS [111], with 1000+ categories).

**Tracking (Deep OC-sort [96]):** The current multi-object tracking model is prone to errors in extremely complex scenes (such as those with numerous occluded or similar objects), which affects DoraemonGPT's ability to locate instances in complex videos accurately.

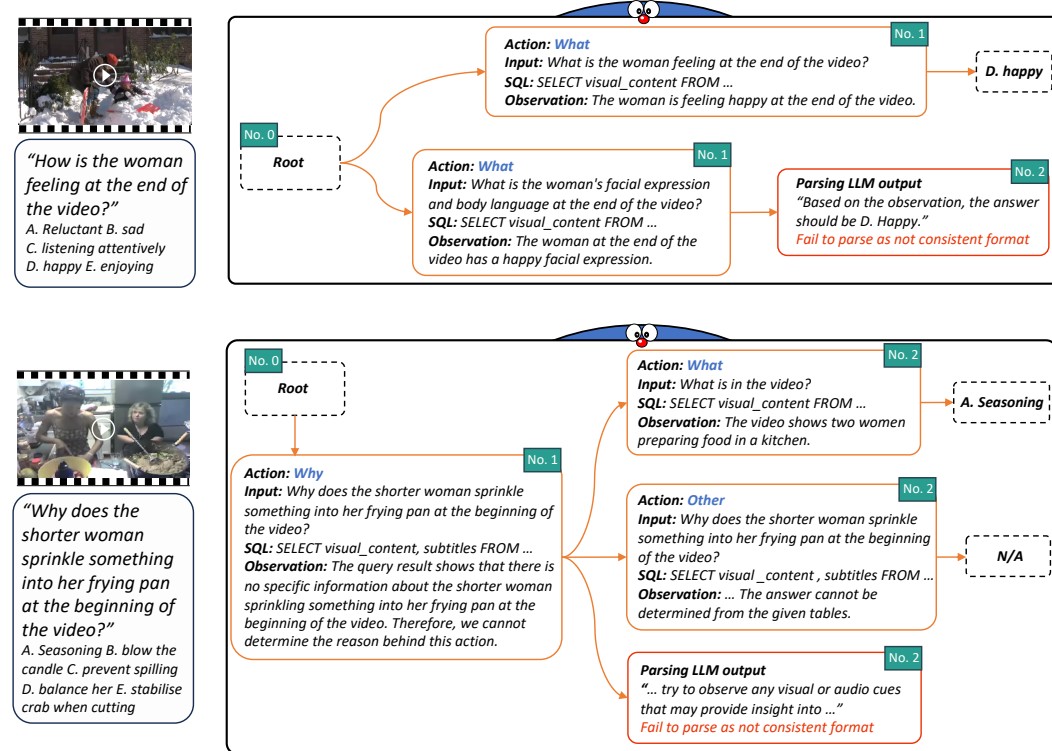

Figure 7: Inference results on NExT-QA [34]. (§A.4)

**Segmentation (YOLOv8-seg [95]):** The segmentation results may not perfectly align with instances' edges, and incomplete segmentation masks can impact the precision of AIGC tools such as video editing (e.g., inpainting).

**Appearance description (BLIP [54]/BLIP-2 [55]):** The textual descriptions cannot accurately capture all the details of an instance (such as intricate clothing details on a human body), which affects DoraemonGPT's handling of tasks related to detailed descriptions.

**Action recognition (InternVideo [97]):** The accuracy is limited by the capabilities of the model, which in turn affects DoraemonGPT's ability to handle action-related inquiries.

**In time-dominant memory:**

**Speech recognition (Whisper [98]):** Current methods can accurately convert audio to text. However, in multi-party conversation scenarios, the methods still cannot accurately perform voiceprint recognition for multiple speakers and accurately separate the results of different speakers. Additionally, it is challenging to match multiple voiceprints with the visual IDs of the speakers. This limitation restricts the ability of DoraemonGPT to infer and deduce the identities of speakers in complex multi-party conversation scenarios, relying solely on the inherent capabilities of LLMs.

**Optical character recognition (OCR [99]):** OCR technology can accurately recognize subtitles and well-structured text. However, it still struggles to robustly handle occluded text and artistic fonts.

**Captioning (BLIP [54]/BLIP-2 [55]/InstructBLIP [100]):** It cannot guarantee that the textual descriptions can accurately cover all the details in the scene, which can affect DoraemonGPT's ability to handle tasks related to detailed descriptions.

Additionally, the domain of the training set for foundation models also affects DoraemonGPT. For instance, currently, visual foundation models trained on real and common scenarios still struggle with extreme lighting conditions or non-realistic scenes (such as simulations or animations).

A.6    EVALUATION ON THE INFERENCE TIME AND TOKEN USAGE EFFICIENCY

For efficiency comparison, we thoroughly analyze the efficiency of DoraemonGPT in comparison with the baselines, ViperGPT and VideoChat. The tables 5 above provide a detailed analysis of the time required for each foundation model used in memory building. When processing videos at a rate of 1 fps, it takes approximately 1 second (or 0.42/0.47s for space/time-dominant memory) to process a 10s video clip using an NVIDIA-A40 GPU. The actual processing time increases linearly with video length.

Table 5: Token Efficiency (Averaged on the NExT-QA [34] s_val).

| Method | Prompt tokens | Node tokens | Steps per Answer | Tokens per Answer | NExT-QA Acc. |
|---|---|---|---|---|---|
| ViperGPT [9] | 4127 | - | - | 4127 | 38.1 |
| VideoChat [35] | 722 | - | - | **722** | 51.0 |
| DoraemonGPT | **617** | 34.6 | 2.3 | 1498 | **54.0** |

In comparison, VideoChat creates a time-stamped memory and takes around 2 seconds to process a 10s video at 1 fps. On the other hand, ViperGPT does not construct a memory but generates a code to invoke foundation models. However, there is a 6.7% chance (60 out of 900 videos) that ViperGPT fails to generate an executable code, and it's difficult to fairly compare the average time of calling foundation models in ViperGPT.

Table 6: Time Analysis of Space-Dominant Memory Construction.

| Model | BLIP-2 [55] | YOLO-v8 [95] | Deep OC-Sort [96] | InternVideo [97] | Sum |
|---|---|---|---|---|---|
| Time(s) | 0.09 | 0.16 | 0.14 | 0.03 | 0.42 |

Due to the influence of simultaneous requests and network delay on ChatGPT's online server, it's impossible to fairly record the run-time of ChatGPT. Thus, a more equitable efficiency comparison when calling ChatGPT is to record the number of tokens used. As shown in the table above, DoraemonGPT's prompt design is more efficient (617 tokens), which is less than VideoChat's approach of directly incorporating video memory into the prompt (722 tokens) and significantly less than ViperGPT's approach of including a large code definition in the prompt (4127 tokens). Additionally, even though the introduction of our MCTS planner divides the task into multiple nodes/steps, DoraemonGPT still requires far fewer tokens on average to obtain an answer compared to ViperGPT (1498 tokens vs 4127 tokens). Furthermore, DoraemonGPT significantly outperform VideoChat (54.0 vs 51.0) on the challenging NExT-QA dataset.

Table 7: Time Analysis of Time-Dominant Memory Construction.

| Model | OCR [99] | Whisper [98] | BLIP-2 [55] | Sum |
|---|---|---|---|---|
| Time(s) | 0.02 | 0.36 | 0.09 | 0.47 |

A.7    QUANTITAVE RESULT ON TVQA+

**Datasets.** The TVQA+ [112] dataset is an enhanced version of the original TVQA [113] dataset, augmented with 310.8K bounding boxes to link visual concepts in questions and answers to depicted objects in videos. It's designed for the spatio-temporal video question answering task, which challenges intelligent systems to identify relevant moments and visual concepts to answer natural language questions about videos. For evaluation, we randomly sample 900 samples from the val set, resulting in a total of 900 questions (s_val).

**Evaluation Metric.** We report accuracy as in NExT-QA [34].

**Performance Comparision.** The results on the TVQA+ [112] confirms again the superiority of DoraemonGPT. From table 8 we can observe that our approach yields remarkable performance, *i.e.*, DoraemonGPT outperforms ViperGPT [9] and VideoChat [35] by **10.2**% and **5.9**%, respectively. In particular, ViperGPT has a 10.9% probability of generating uncompilable code (98 out of 900 videos). However, even when filtering out these failures, its performance (30.1%) is still lower compared to VideoChat and DoraemonGPT, which are specifically designed for dynamic videos. This is consistent with the findings on NExT-QA [34].

Table 8: Comparison of our DoraemonGPT with SOTAs on TVQA+ [112]. †: reimplement using the officially released codes. ∗: we filter out those failed executions (*i.e.*, compilation error) of ViperGPT [9] and record the performance on successful executions (802/900 on s_val).

| Method | Split | Accuracy |
|---|---|---|
| †ViperGPT [9] | s_val | 26.8 |
| ∗†ViperGPT [9] | s_val | 30.1 |
| †VideoChat [35] | s_val | 34.4 |
| DoraemonGPT | s_val | **40.3** |

## A.8 LIMITATIONS

Despite its comprehensive and conceptually elegant system, DoraemonGPT has some limitations for future studies. First, although TSM is a simple and effective way to decouple and handle spatial-temporal reasoning and DoraemonGPT has shown effectiveness with two task-related memory types (*space-dominant* and *time-dominant*), we believe that by further subdividing the types of tasks, we can introduce more nuanced categories of memory (*e.g.*, human-centric memory) to construct task-related information with greater task-relevance. However, at present, the design of memory types is still a heuristic and manually driven process, lacking an automated design method. Second, the establishment of memory relies on the available foundation models (*e.g.*, BLIP-2 [55]). In other words, foundation models' performance directly influences memory's reliability. Incorrect model predictions will introduce noise into the memory, thereby reducing its reliability and affecting the accuracy of decision-making. Additionally, foundation models may struggle to effectively extract the required video attributes in real-world scenarios that are difficult to generalize (*e.g.*, low light, blurriness, occlusions, *etc.*). Third, the accuracy of planning in DoraemonGPT is limited by the capabilities of LLMs. When using a small-scale or insufficiently trained LLM, the likelihood of DoraemonGPT exploring reasonable solutions may be significantly reduced. Last, while the MCTS planner significantly improves the decision making ability of DoraemonGPT, it also introduces additional computational cost. This means that DoraemonGPT may only be available on high-end computing systems or online LLM services [1], limiting its use in real-time, resource-constrained scenarios.

## A.9 BROADER IMPACTS

DoraemonGPT aims to solve real-world dynamic tasks with LLMs and can handle video-based reasoning tasks, potentially revolutionizing several fields. Our system has potential applications in autonomous vehicles, surveillance systems, and interactive robotics, where dynamic understanding and decision making are crucial. However, it is important to consider the ethical implications and potential misuse of such systems. **First**, like many AI systems, DoraemonGPT could be exploited by malicious individuals for video manipulation or generating misleading content, posing threats to privacy and security. Protecting against such potential misuse requires robust safeguards and measures to detect and prevent malicious activities. **Second**, biases in the training data of LLMs or foundation models could unintentionally perpetuate discriminatory behavior. Mitigating biases and promoting fairness in the training and deployment of DoraemonGPT is essential to ensure equitable outcomes. **Third**, the reliance on external knowledge sources highlights the importance of data access and usage rights. Users and developers must adhere to regulations and ethical guidelines associated with these resources to avoid any legal complications. **Fourth**, the methodology introduced in DoraemonGPT holds potential for application of LLM-driven agents beyond the realm of vision. The rapid expansion of LLM-driven agents opens doors to transformative impacts across various fields [20, 67–72]. DoraemonGPT, with its novel approach to modeling the dynamic aspects of visual scenes, tackles complex tasks through a computer vision lens. This innovation could extend its influence to other domains. For instance, in tool usage, our MCTS planner can offer effective exploration strategies in large solution spaces. Additionally, when it comes to open-world environments, our symbolic memory could provide precise guidances through symbolic language. This is particularly relevant for interactive planning scenarios[67, 69].

