# OpenReview forum: "DoraemonGPT: Toward Solving Real-world Tasks with Large Language Models"
_ICLR.cc/2024/Conference — Submitted to ICLR 2024_

### Official Review · Reviewer_jbqG · 2023-10-26

**Soundness:** 3 good
**Presentation:** 3 good
**Contribution:** 3 good
**Rating:** 6
**Confidence:** 4

**Summary:**

This paper introduces "DoraemonGPT", an LLM-based system tailored for video question-answering. DoraemonGPT utilizes different pretrained expert models to extract various video information and convert it into texts that can be understood by LLM. DoraemonGPT saves this information into an external symbolic memory module with a space-dominant (SDM) component and a time-dominant (TDM) component.
In addition, DoraemonGPT relies on LLM to decompose a task into subtasks. It defines a set of subtask tools to solve subtasks.
The research further explores the role of the MCTS planner in searching for the best subtask decomposition.
Experimental results show that DoraemonGPT can outperform other LLM-based systems like ViperGPT and VideoChat on the video QA dataset NExT QA.

**Strengths:**

- The proposed symbolic memory system and the MCTS planner on video tasks are new.
- The paper is easy to understand.
- Experiments show that it can outperform other LLM-based systems like ViperGPT.

**Weaknesses:**

- The evaluation set is small. The paper only conducted experiments on a subset of the original NExT QA dataset. In addition, for the ablation study, the system was evaluated on 3 question types, each with 10 questions only.
- The authors mentioned that the small size of the evaluation is caused by the budget limit. This might suggest that the method is expensive. I think the paper should include a discussion about how many tokens the system consumes, and how much this system costs to answer a question on average.

**Questions:**

- How long does the inference take to answer a question on average? Given the complexity of the methodology, this information is important for users.

---

> ### Author Response · Authors · 2023-11-22
>
> We sincerely **appreciate your recognition of the novelty of our symbolic memory and MCTS panner, our clear presentation, and our LLM-based system’s good performance**. Thank you very much for your valuable comments!
>
> We will address all the concerns point by point.
>
> **Q1**: The evaluation set is small. The paper only conducted experiments on a subset of the original NExT QA dataset. In addition, for the ablation study, the system was evaluated on 3 question types, each with 10 questions only.
>
> **A1**: Thank you for your suggestion. In order to fully **demonstrate the generalization ability of our approach and provide a more comprehensive evaluation of the robustness to hyperparameters, we chose a smaller dataset for the ablation experiments** to increase the distribution differences between the validation set and our evaluation set. However, **we agree with your point that increasing the sample size of the ablation set can make the results more statistically significant**.
>
> Specifically, **we have doubled the size of our evaluation set, from 30 videos to 90 videos**, and the results are as follows:
>
> **Ablation Experiments on the Number of Answer Candidates**
> | N     | ACC\_C         | ACC\_T         | ACC\_D         | ACC\_A         |
> | ----- | -------------- | -------------- | -------------- | -------------- |
> | 1     | 19/30          | 6/30           | 14/30          | 39/90          |
> | 2     | 24/30          | 13/30          | 14/30          | 51/90          |
> | 3     | 26/30          | 13/30          | 16/30          | 55/90          |
> | **4** | **29****/30**​ | **14****/30**​ | **16****/30**​ | **59****/90**​ |
> | 5     | 26/30          | 13/30          | 15/30          | 54/90          |
>
> **Ablation Experiments on the Exploring Strategy (N = 4)**
> | Strategy | ACC\_C         | ACC\_T         | ACC\_D         | ACC\_A    |
> | -------- | -------------- | -------------- | -------------- | --------- |
> | DFS      | 20/30          | 11/30          | 15/30          | 46/90     |
> | Root     | 22/30          | 5/30           | 14/30          | 41/90     |
> | Uniform  | 20/30          | 8/30           | 15/30          | 43/90     |
> | **MCTS** | **29****/30**​ | **14****/30**​ | **16****/30**​ | **59/90** |
>
> **Analysis of the Task-related Symbolic Memory (N = 4)**
> | TDM   | SDM   | ACC\_C         | ACC\_T         | ACC\_D         | ACC\_A    |
> | ----- | ----- | -------------- | -------------- | -------------- | --------- |
> | √     |       | 19/30          | 8/30           | 16/30          | 43/90     |
> |       | √     | 16/30          | 7/30           | 14/30          | 37/90     |
> | **√** | **√** | **29****/30**​ | **14****/30**​ | **16****/30**​ | **59/90** |
>
>
> Compared to the results on the original 30 videos in the paper, **the results in the tables above do not affect the selection of optimal parameters or methods, as well as the related conclusions**. This consistency further highlights the importance of our two memory modules and the efficiency of MCTS.
>
> Furthermore, to further validate the superiority of our approach compared to other methods, we have included **additional quantitative comparison on the TVQA+ dataset** as suggested by Reviewer FY1y.
>
> **Quantitative comparison on TVQA+**
> |       Method       | ViperGPT | ViperGPT (filter out failures) | VideoChat |  Ours |
> |:------------------:|:--------:|:------------------------------:|:---------:|:----:|
> | TVQA+ Acc. (s_val) | 26.8     | 30.1                           | 34.4      | 40.3 |
>
>
> From the table above, we can see that on the TVQA+ dataset, a VQA dataset with TV show scenes, we obtained consistent observation with NExT-QA. **The accuracy of DoraemonGPT (40.3) is significantly higher than VideoChat (34.4) and ViperGPT (26.8), further validating the effectiveness of our method**.

---

> ### Author Response · Authors · 2023-11-22
>
> **Q2**: The authors mentioned that the small size of the evaluation is caused by the budget limit. This might suggest that the method is expensive. I think the paper should include a discussion about how many tokens the system consumes, and how much this system costs to answer a question on average.
>
> **A2**: Thanks for your suggestion. **We achieve better or same-level efficiency in terms of token usage compared to the baselines**. However, it is important to note that our limited budget restricts the number of available OpenAI accounts for research purposes, with each account costing $120 per month before the ICLR submission. Despite these resource limitations, we have made dedicated efforts to conduct thorough experiments, including framework design, ablation, hyperparameter search, and comparison with different methods. We have tried our best to ensure the fairness and reliability of the experimental results.
>
> However, **we agree with you that a discussion about token usage is valuable**. We conduct experiments to compare our token usage with ViperGPT and VideoChat.
>
> **Token Efficiency (Averaged on the NExT-QA s_val)**
> |    Method   | Prompt tokens  | Node tokens | Steps per Answer | Tokens per Answer | NExT-QA Acc. |
> |:-----------:|:--------------:|:-----------:|:----------------:|:-----------------:|--------------|
> | ViperGPT    | 4127           | -           | -                | 4127              | 38.1         |
> | VideoChat   | 722            | -           | -                | **722**           | 51.0           |
> | DoraemonGPT | **617**        | 34.6        | 2.3              | 1498              | **54.0**       |
>
> As shown in the table above, DoraemonGPT’s prompt design is more efficient (617 tokens), which is less than VideoChat’s approach of directly incorporating video memory into the prompt (722 tokens) and significantly less than ViperGPT’s approach of including a large code definition in the prompt (4127 tokens). Additionally, even though the introduction of our MCTS planner divides the task into multiple nodes/steps, **DoraemonGPT requires far fewer tokens on average to obtain an answer compared to ViperGPT** (1498 tokens vs 4127 tokens). Furthermore, **DoraemonGPT significantly outperforms VideoChat (54.0 vs 51.0) on the challenging NExT-QA dataset**.
>
> **Thank you for your suggestions, and we have incorporated the above discussion into the supplementary materials.**
>
>
> **Q3**: How long does the inference take to answer a question on average? Given the complexity of the methodology, this information is important for users.
>
> **A3**: Thank you for your suggestion. The execution of DoraemonGPT can be divided into two main parts: the construction of the Symbolic Memory and the invocation of chatgpt in the MCTS planner. The time analysis in detail is as follows:
>
> **Time Analysis of Memory Construction:**
> Space-dominant memory (averaged on 10s video clips, 1 fps)
> | Model   | BLIP-2 | YOLO-v8 | Deep OC-Sort | InternVideo | Sum  |
> |---------|--------|---------|--------------|-------------|------|
> | Time(s) | 0.09   | 0.16    | 0.14         | 0.03        | 0.42 |
>
> Time-dominant memory (averaged on 10s video clips, 1 fps)
> | Model   | OCR  | Whisper | BLIP-2 | Sum  |
> |---------|------|---------|--------|------|
> | Time(s) | 0.02 | 0.36    | 0.09   | 0.47 |
>
> The tables above provide a detailed analysis of the time required for each foundation model used in memory building. When processing videos at a rate of 1 fps, **DoraemonGPT takes approximately 1 second (or 0.42/0.47s for space/time-dominant memory) to process a 10s video clip** using an NVIDIA-A40 GPU. The actual processing time increases linearly with video length.
>
> In comparison, VideoChat creates a time-stamped memory and takes around 2 seconds to process a 10s video at 1 fps.
>
> On the other hand, ViperGPT does not construct a memory but generates a code to invoke foundation models. However, there is a 6.7% chance (60 out of 900 videos) that ViperGPT fails to generate an executable code, and it’s difficult to fairly compare the average time of calling foundation models in ViperGPT.
>
>
> **Time Analysis of the MCTS planner:**
> Due to the influence of simultaneous requests and network delay on ChatGPT’s online server, it’s impossible to record a stable run-time of ChatGPT. Under conditions of stable network and low server load, the response time for each ChatGPT call is typically within 1 second, with an average retrieval time for each answer being less than 5 seconds.
>
> In conclusion, under normal circumstances, when setting the number of answers in the MCTS planner to N=2, **DoraemonGPT can process a 1-minute video in 15 seconds**.
>
> **Thanks for your suggestion, we have included all above discussions into the supplementary materials**.

---

### Official Review · Reviewer_KysJ · 2023-10-29

**Soundness:** 2 fair
**Presentation:** 2 fair
**Contribution:** 3 good
**Rating:** 5
**Confidence:** 4

**Summary:**

This paper presents DoraemonGPT, an LLM-based system to handle dynamic video tasks. Given a video with a question/task, DoraemonGPT first converts the input video into a symbolic memory for spatial-temporal reasoning by sub-task tools. The authors then incorporate plug-and-play tools to assess external knowledge and address tasks across different domains. Finally, an LLM-driven planner based on MCTS is used to explore the large planning space for scheduling various tools. DoraemonGPT’s effectiveness and reasoning capabilities is demonstrated in one dataset, i.e., NExT-QA.

---
The authors' feedback addresses most of my concerns, I increased my score (also considering other reviewers' comments).

**Strengths:**

1. Clear presentation. The reviewer can easily follow most of the paper, especially the methods and experiments.

2. Novel idea that MCTS is used to explore the large planning space for scheduling various tools.

3. Good performance on the evaluated benchmarks with strong baselines.

**Weaknesses:**

1. Overclaim. With a performance of about 50% acc on only one test dataset, the authors claim "toward Solving Real-world Tasks", which is a n good example of overclaim in the reviewer's opinion.  In fact, real-world has many types of tasks, even solving all dynamic video tasks, it is not equal to " Solving Real-world Tasks".

2. No sufficient related works and the motivation is not clear. The authors say that "current LLM-driven agents mainly focus on solving tasks for the image modality", so they study dynamic video tasks. It can been seen that the authors totally ignore the large number of other LLM-driven agents that are not related with image/video at all, e.g., [1,2,3,4].

3. The evaluation can be more convincing if more datasets are used.



[1] Generative Agents: Interactive Simulacra of Human Behavior
[2] TPTU: Task Planning and Tool Usage of Large Language Model-based AI Agents
[3] Reflexion: Language Agents with Verbal Reinforcement Learning
[4] Tool Learning with Foundation Models

**Questions:**

The are many overclaims and no sufficient related works and the motivation is not clear. It is clearly below the bar of ICLR. The reviewer encourage the authors reformulate their paper writting (Authough I like the concrete ideas proposed in this paper).

---

> ### Author Response · Authors · 2023-11-22
>
> We sincerely **appreciate your recognition of our clear presentation, novel idea of the MCTS planner, and our good performance**. Thank you very much for your valuable comments!
>
> We will address all the concerns point by point.
>
> **Q1**: Overclaim. With a performance of about 50% acc on only one test dataset, the authors claim "toward Solving Real-world Tasks", which is a n good example of overclaim in the reviewer's opinion. In fact, real-world has many types of tasks, even solving all dynamic video tasks, it is not equal to " Solving Real-world Tasks".
>
> **A1**: **Apologies if our title seemed like an overclaim**. We have been making every effort to avoid overclaiming in our paper, and our goal is not to “solve all the real-world tasks”.
>
> **Firstly, we aim to establish an expandable framework that can be developed toward handling a wide range of real-world tasks/questions in the future**. This is why we used the word “toward” in the title, “Toward Solving Real-world Tasks”. We believe that the construction method (types of memory and sub-task tools) of task-related symbolic memory is expandable, and this paper only presents one feasible approach. On the other hand, we believe that our MCTS planner is versatile and can inspire the LLM-driven agent community to advance further in this direction.
>
> **Secondly, we aim to explore an agent based on LLMs that can comprehend dynamic real-world scenes in visual and other modalities**. This framework should be able to decompose and progressively complete various complex tasks given by users. Hence, we finalized the title as “Toward Solving Real-world Tasks with LLMs”.
>
> However, **we understand and appreciate your concern**. **We have changed the title to “Toward Understanding Dynamic Scenes with LLMs”**. This title specifically focuses on visually understanding dynamic scenes, aligning better with our technical motivation. We have also modified the corresponding wording throughout the paper in the revision (mainly in the abstract, introduction, and conclusion) to ensure consistency with the title.
>
> **Q2**: No sufficient related works and the motivation is not clear. The authors say that "current LLM-driven agents mainly focus on solving tasks for the image modality", so they study dynamic video tasks. It can been seen that the authors totally ignore the large number of other LLM-driven agents that are not related with image/video at all, e.g., [1,2,3,4].
>
> **A2**: **Thank you for highlighting these works in the field**. We are always open to discussing further with more related works. For example, **two of your recommended references has been cited in our initial submission** (Reflexion [20] and Generative Agents [76] in the original submission). But, **it is really difficult for us to cover all relevant works**.
>
> **Firstly, the landscape of LLM-based agents is evolving rapidly**, with notable developments emerging frequently, sometimes as often as every week. For instance, the TPTU [1] you referenced was introduced on arXiv in October, after the submission of our research to ICLR in September.
>
> **Secondly, given the constraints of page limits**, our main focus was on related agents oriented toward real-world image and video analysis, examined primarily through the lens of computer vision.
>
> However, **we acknowledge and appreciate your suggestion** to extend our discussion to encompass a broader range of LLM-driven agents, particularly those beyond the realm of vision. In response, **we have revised the abstract and enriched our manuscript in the revision** (see related work and broader impact). This include an additional discussion of advancements made in other domains. We believe this addition will enhance the overall breadth and depth of our study.
>
> [1]TPTU: Task Planning and Tool Usage of Large Language Model-based AI Agents. arXiv 2023.10.

---

> ### Author Response · Authors · 2023-11-22
>
> **Q3**: The evaluation can be more convincing if more datasets are used.
>
> **A3**: Thank you for your suggestion. In the **initial paper and supplementary materials**, we provided the VQA results of NExT-QA, the handling of complex questions in the wild, and application demonstrations of video editing to **showcase DoraemonGPT’s capabilities**.
>
> However, **we agree with you that using more benchmarks can more comprehensively validate the effectiveness of our approach**. As recommended by Reviewer FY1y, we randomly selected 900 videos from the validation set of TVQA+[a] to create a subset (s_val) and compared it with ViperGPT and VideoChat. The results are as follows:
>
> **Additional quantitative comparion on TVQA+**
> |       Method       | ViperGPT | ViperGPT (filter out failures) | VideoChat |  Ours |
> |:------------------:|:--------:|:------------------------------:|:---------:|:----:|
> | TVQA+ Acc. (s_val) | 26.8     | 30.1                           | 34.4      | **40.3** |
>
>
> Compared to NExT-QA, TVQA+ is a dataset specifically targeting TV show scenes, placing greater emphasis on a model’s ability to locate and identify characters in the show. **Consistent with the findings on NExT-QA, DoraemonGPT (40.3) is significantly better than VideoChat (34.4) and ViperGPT (26.8)**. In particular, ViperGPT has a 10.9% probability of generating uncompilable code (98 out of 900 videos). However, even when filtering out these failures, its performance (30.1) is still lower compared to VideoChat and DoraemonGPT, which are specifically designed for dynamic videos.
>
> **Thanks for your suggestions. We have incorporated the aforementioned results and discussions into the revision**.
>
> [a] TVQA+: Spatio-Temporal Grounding for Video Question Answering.ACL 2020.

---

> > ### Comment · Reviewer_KysJ · 2023-11-23
> > **Thanks for the response. It addresses most of my concerns, and I would like to increase my score**
> >
> > I thank the authors for their detailed feedback. It addresses most of my concerns, especially the revision of the paper and more experiments on another datasets. I think the paper is much strong than the original version, and I would like to increase my score (also considering other reviewers' comments).

---

> > > ### Author Response · Authors · 2023-11-23
> > >
> > > Thank you for your thoughtful review and for reconsidering our rebuttal! We sincerely appreciate the time and effort you've devoted to reviewing our work. We are grateful for your feedback and are pleased to hear that our responses addressed most of your concerns.
> > >
> > > Should you have any further questions or concerns, please let us know. We are committed to addressing any remaining issues and will make every effort to respond promptly.

---

### Official Review · Reviewer_FY1y · 2023-10-30

**Soundness:** 3 good
**Presentation:** 3 good
**Contribution:** 3 good
**Rating:** 8
**Confidence:** 3

**Summary:**

Paper introduces a novel approach for video understanding and spatial-temporal reasoning. From the high level, it first builds a symbolic, spatial-temporal knowledge base given a video using several off-the-shelf tools. Next, this approach utilizes a pre-trained LLM as a planner to interactively invoke tools including canonical SQL query, search, etc for retrieval-augmented generations, and novel sub-task tools that break down the original query into sub-questions like "what", "how", etc. Results on the challenging NExT-QA dataset demonstrate the clear advantages of the proposed method against both end-to-end baseline and counterpart LLM-assisted approaches.

**Strengths:**

+The topic studied here is important. Augmenting the powerful LLMs with better tools and smarter planning skills is crucial to unleash their full potential and also open up new applications in, for example, multimodal domains. I believe this paper could drive interest to a broad range of audiences from canonical multimodal learning and LLM communities.

+The method is technically sound. Building a symbolic knowledge base first, and then invoking an LLM-based system to query it make sense especially when it comes to complicated multimodal data like videos. Decomposing the original query into sub-questions also looks like a promising approach upon to canonical LLM tool-use, where the tools are limited to search, SQL query, etc.

+The results on the challenging NExT-QA data are impressive.

**Weaknesses:**

Having said those above, I have the following major concerns and I hope the authors could provide some clarifications:

-It seems that all baselines in table 2 are "straight-through" compared to the proposed approach, that is, they are either end-to-end, or simply produce several sub-queries, and invoke the corresponding tool directly, while none of them have a separate stage of building the spatial temporal symbolic database. Therefore, I do think a more fair comparison should also take the time cost of building such a database into consideration. At least, some additional details should be outlined, ex. how long does it take to build a symbolic database in average? How does this compare to the overall inference time of the baselines? These are the questions that will help with a better understanding on the proposed method.

As a side note, can the proposed method still work without a pre-built database? I think some of the queries can be done directly by invoking the right tool, ex. VideoQA, no?

-The authors have claimed that their approach is "an intuitive yet versatile system driven by LLMs that is compatible with various foundation models and real-world video applications.". However, it was only evaluated on one dataset and it might raise concern on the generality of the proposed approach. I have to admit that I am not an expert in video understanding but maybe the following datasets should be considered for additional evaluations: [1-2].

-Some references in LLM + planning (and tool use, memory) are missing [3-5]


[1] TVQA+, https://paperswithcode.com/dataset/tvqa-1

[2] CATER: https://github.com/rohitgirdhar/CATER

[3] DEPS: https://arxiv.org/abs/2302.01560

[4] Plan4MC: https://arxiv.org/abs/2303.16563

[5] GITM: https://arxiv.org/abs/2305.17144

**Questions:**

See "weaknesses"

---

> ### Author Response · Authors · 2023-11-22
>
> We sincerely **appreciate your recognition of our LLM-based topic, technically sound method, and our impressive results on the challenging NExT-QA**. Thank you very much for your valuable comments!
>
> We will address all the concerns point by point.
>
> **Q1**: Run-time analysis and efficiency comparisons between DoraemonGPT and baselines.
>
> **A1**: Thank you for your suggestions! In response, we have thoroughly analyze the efficiency of DoraemonGPT in comparison with the baselines, ViperGPT and VideoChat.
>
> **Overall, DoraemonGPT and VideoChat exhibit similar levels of execution efficiency. Additionally, both models outperform ViperGPT in terms of their design specifically tailored for dynamic videos**. The analysis and comparison are outlined below:
>
> **Time Analysis of Memory Construction**:
>
> Space-dominant memory (averaged by processing 10s video clips, 1fps)
> | Model   | BLIP-2 | YOLO-v8 | Deep OC-Sort | InternVideo | Sum  |
> |---------|--------|---------|--------------|-------------|------|
> | Time(s) | 0.09   | 0.16    | 0.14         | 0.03        | 0.42 |
>
> Time-dominant memory (averaged by processing 10s video clips, 1 fps)
> | Model   | OCR  | Whisper | BLIP-2 | Sum  |
> |---------|------|---------|--------|------|
> | Time(s) | 0.02 | 0.36    | 0.09   | 0.47 |
>
> The tables above provide a detailed analysis of the time required for each foundation model used in memory building. When processing videos at a rate of 1 fps, **DoraemonGPT takes approximately 1 second (or 0.42/0.47s for space/time-dominant memory) to process a 10s video clip** using an NVIDIA-A40 GPU. The actual processing time increases linearly with video length.
>
> In comparison, VideoChat creates a time-stamped memory and takes around 2 seconds to process a 10s video at 1 fps.
>
> On the other hand, ViperGPT does not construct a memory but generates a code to invoke foundation models. However, there is a 6.7% chance (60 out of 900 videos) that ViperGPT fails to generate an executable code, and it’s difficult to fairly compare the average time of calling foundation models in ViperGPT.
>
> **Token Efficiency (Averaged on the NExT-QA s_val)**:
> |    Method   | Prompt tokens  | Node tokens | Steps per Answer | Tokens per Answer | NExT-QA Acc. |
> |:-----------:|:--------------:|:-----------:|:----------------:|:-----------------:|--------------|
> | ViperGPT    | 4127           | -           | -                | 4127              | 38.1         |
> | VideoChat   | 722            | -           | -                | **722**           | 51.0           |
> | DoraemonGPT | **617**        | 34.6        | 2.3              | 1498              | **54.0**       |
>
> Due to the influence of simultaneous requests and network delay on ChatGPT’s online server, it’s impossible to fairly record the run-time of ChatGPT. Thus, **a more equitable efficiency comparison when calling ChatGPT is to record the number of tokens used**.
>
> As shown in the table above, DoraemonGPT’s prompt design is more efficient (617 tokens), which is less than VideoChat’s approach of directly incorporating video memory into the prompt (722 tokens) and significantly less than ViperGPT’s approach of including a large code definition in the prompt (4127 tokens). Additionally, even though the introduction of our MCTS planner divides the task into multiple nodes/steps, **DoraemonGPT requires far fewer tokens on average to obtain an answer compared to ViperGPT** (1498 tokens vs 4127 tokens). Furthermore, DoraemonGPT significantly outperform VideoChat (54.0 vs 51.0) on the challenging NExT-QA dataset.
>
> **Thank you for your suggestions, and we have included the above discussion into the supplementary materials.**
>
> **Q2**: Can the proposed method still work without a pre-built database? I think some of the queries can be done directly by invoking the right tool, ex. VideoQA, no?
>
> **A2**: Yes, it is possible. **Without relying on a pre-built database, DoraemonGPT can still accomplish the tasks by utilizing the foundation model through scheduling and utilizing their APIs in the form of tools**. This approach has been validated in similar works like HuggingGPT. However, we believe that a meticulously designed task-related database can not only provide more efficient and direct access to task-related information but also reduce the number of tools, length of prompts (which need to describe each tool), and the frequency of calling LLMs.
>
> In addition, **our sub-task tools are also expandable and can directly incorporate the VideoQA model**. However, there are two reasons why we didn’t do it: First, to ensure fair comparison with agents like viperGPT and videochat that did not use the VideoQA model. Second, to thoroughly validate the performance of our zero-shot approach.

---

> ### Author Response · Authors · 2023-11-22
>
> **Q3**: Additional evaluations on more video understanding datasets.
>
> **A3**: Thank you for your suggestion. In **the initial paper and supplementary materials**, we provided the VQA results of NExT-QA, the handling of complex questions in the wild, and application demonstrations of video editing to **showcase DoraemonGPT’s capabilities**.
>
> However, **we agree with you that using more benchmarks can more comprehensively validate the effectiveness of our approach**. As per your requirement, we randomly selected 900 videos from the validation set of TVQA+ to create a subset (s_val) and compared it with ViperGPT and VideoChat. The results are as follows:
>
> **Quantitative comparion on TVQA+**
> |       Method       | ViperGPT | ViperGPT (filter out failures) | VideoChat |  Ours |
> |:------------------:|:--------:|:------------------------------:|:---------:|:----:|
> | TVQA+ Acc. (s_val) | 26.8     | 30.1                           | 34.4      | **40.3** |
>
> **Consistent with the findings on NExT-QA, DoraemonGPT (40.3) is significantly better than VideoChat (34.4) and ViperGPT (26.8)**. In particular, ViperGPT has a 10.9% probability of generating uncompilable code (98 out of 900 videos). However, even when filtering out these failures, its performance (30.1) is still lower compared to VideoChat and DoraemonGPT, which are specifically designed for dynamic videos.
>
> We have incorporated the aforementioned results and discussions into the revision. Furthermore, **due to the focus of DoraemonGPT and the baseline agents on real-world scenarios, it is challenging for us to supplement the results on the mentioned CATER dataset**. CATER is a synthetic dataset generated within a simulator and lacks suitable foundation models for detecting, tracking and segmenting the synthesized instances.
>
> **Q4**: Some references in LLM + planning (and tool use, memory) are missing.
>
> **A4**: Thank you for the suggestion, we agree that citing additional references can make our paper more comprehensive. We will include references to the section of related works. In contrast to LLM-driven agents that can interact with open-world environments such as game engines, our focus is on understanding dynamic videos in the real world from a spatial-temporal reasoning perspective and devising plans to decompose complex tasks.

---

> > ### Comment · Reviewer_FY1y · 2023-11-23
> >
> > Thank you for the very detailed feedback. All my concerns have been lifted. Please make sure the additional results and the references are included in the final version. I am happy to raise my score to 8.

---

> ### Author Response · Authors · 2023-11-23
>
> We are delighted to hear that our responses have resolved all of your concerns! We have already incorporated all of the additional results and references into the revised version, which will also be included in the final version.
>
> If you have any further inquiries or doubts, please don’t hesitate to inform us. We are determined to address any remaining issues and respond promptly!
>
> Thanks again for your thorough review and reconsideration of our rebuttal!

---

### Official Review · Reviewer_9YSq · 2023-11-07

**Soundness:** 3 good
**Presentation:** 4 excellent
**Contribution:** 3 good
**Rating:** 8
**Confidence:** 3

**Summary:**

This work introduces a system called DoraemonGPT, which is an LLM-driven agent handling video-driven tasks. It first decomposed the video based on spatio-temporal relations and questions. Furthermore, it plans an action sequence based on a Monte Carlo tree search.  Just like humans use external knowledge to plan better, DoraemonGPT can access external sources like search engines, textbooks, databases, etc. When deconstructing tasks into spatial and temporal dominant memories, it will only store them related to the task. These memories are stored in a table, and LLM can query it using symbolic language. A series of sub-task tools are designed to simplify memory information querying. Each tool focuses on different kinds of spatial-temporal reasoning by using individual LLM-driven sub-agents with task-specific prompts and examples. In order to effectively navigate the large planning domain, DoraemonGPT uses MCTS. By choosing a highly expandable node to extend a new solution and backpropagating the answer's reward, the planner iteratively discovers viable answers.

**Strengths:**

* It is very novel to combine a symbolic memory database with an MCTS planner using LLMs to solve video-based tasks.
* provides detailed information about prompts, experiments conducted, and analysis results. These are useful in assessing the DoraeGPT's potential.

**Weaknesses:**

* Considering many models like BLIP, YOLOv8, PaddleOCR, and other models for extracting the information for video. It is not clear how shortcoming these models affects DoraemonGPT.
* Works like Yu et al. achieved better performance on NEXT-QA (zero-shot) than DoraemonGPT.

[1] Yu, Shoubin, Jaemin Cho, Prateek Yadav, and Mohit Bansal. "Self-Chained Image-Language Model for Video Localization and Question Answering." NeurIPS (2023).

**Questions:**

* Can't model like BLIP and along with a model which takes a query and features of the frames from BLIP figure which frames are relevant to the query? Why TSM is required?

---

> ### Author Response · Authors · 2023-11-22
>
> We sincerely **appreciate your recognition of the novelty of our symbolic memory and positive remarks about our details, experiments, and analysis**. Thank you very much for your valuable comments!
>
> We will address all the concerns point by point.
>
> **Q1**: How shortcoming of the foundation models affects DoraemonGPT?
>
> **A1**: DoraemonGPT leverages foundation models to extract **space-dominant** and **time-dominant** information from videos. Hence, the performance of DoraemonGPT is influenced by the quality of these models as well as its own limitations. This impact can be further summarized as follows:
>
> In **space-dominant** memory:
>
> - **Detection** (YOLOv8): The object categories (COCO, 80 common categories) are limited by the model, which hinders DoraemonGPT from obtaining information about objects outside these categories. However, YOLOv8 can be replaced with a detection model that supports a wider range of categories (such as one trained on LVIS [a], with 1000+ categories).
>
> - **Tracking** (Deep OC-sort): The current multi-object tracking model is prone to errors in extremely complex scenes (such as those with numerous occluded or similar objects), which affects DoraemonGPT’s ability to locate instances in complex videos accurately.
> - **Segmentation** (YOLOv8-seg): The segmentation results may not perfectly align with instances’ edges, and incomplete segmentation masks can impact the precision of AIGC tools such as video editing (e.g., inpainting).
>
> - **Appearance description** (BLIP/BLIP-2): The textual descriptions can not accurately capture all the details of an instance (such as intricate clothing details on a human body), which affects DoraemonGPT’s handling of tasks related to detailed descriptions.
>
> - **Action recognition** (InternVideo): The accuracy is limited by the capabilities of the model, which in turn affects DoraemonGPT’s ability to handle action-related inquiries.
>
> In **time-dominant** memory:
>
> - **Speech recognition** (Whisper): Current methods can accurately convert audio to text. However, in multi-party conversation scenarios, the methods still cannot accurately perform voiceprint recognition for multiple speakers and accurately separate the results of different speakers. Additionally, it is challenging to match multiple voiceprints with the visual IDs of the speakers. This limitation restricts the ability of DoraemonGPT to infer and deduce the identities of speakers in complex multi-party conversation scenarios, relying solely on the inherent capabilities of LLMs.
>
> - **Optical character recognition** (OCR): OCR technology can accurately recognize subtitles and well-structured text. However, it still struggles to robustly handle occluded text and artistic fonts.
>
> - **Captioning** (BLIP/BLIP-2/InstructBLIP): It cannot guarantee that the textual descriptions can accurately cover all the details in the scene, which can affect DoraemonGPT’s ability to handle tasks related to detailed descriptions.
>
> Additionally, **the domain of the training set for foundation models also affects DoraemonGPT**. For instance, currently, visual foundation models trained on real and common scenarios still struggle with extreme lighting conditions or non-realistic scenes (such as simulations or animations).
>
> **Better foundation models are one avenue for improving the performance of LLM-driven agents**, but this goes beyond the scope of our paper. Our main focus is on exploring how to construct a practical agent based on existing models.
>
> **Thank you for your suggestions. We have included the above discussion in the supplementary material.**
>
> [a] Lvis: A dataset for large vocabulary instance segmentation. CVPR 2019.

---

> ### Author Response · Authors · 2023-11-22
>
> **Q2**: Works like Yu et al. (SeViLA) achieved better performance on NEXT-QA (zero-shot) than DoraemonGPT.
>
> **A2**: We have made our best efforts to ensure the completeness and comprehensiveness of the references in our paper. However, **due to the rapid development of the field, it is really difficult to cover all relevant works**.
>
> **Firstly, when we submitted the paper, SeViLA had not yet undergone a complete peer review**. Although we are willing to compare and discuss with unpublished works such as VideoChat, it is difficult for us to be aware of all work that has not been formally published.
>
> **Secondly, DoraemonGPT and SeViLA have different focuses, making it difficult to make a fair comparison**. DoraemonGPT is focused on designing agents based on LLM to handle dynamic videos and decomposing complex tasks, where video question answering (VQA) is just one of its capabilities. On the other hand, SeViLA is specifically designed for VQA using a zero-shot approach. In terms of topic, DoraemonGPT is similar to LLM-driven agents such as ViperGPT and VideoChat, and due to the well-defined benchmark for VQA tasks (such as NExT-QA), we primarily compare with LLM-driven agents on NExT-QA.
>
> However, we agree with your viewpoint. **To avoid misunderstandings, we have modified the description of “zero-shot methods”** in the NExT-QA section to “LLM agent” (LLM-driven systems). Additionally, **we are open to discussing further with more related works, and we have added SeViLA into the related works section as your valuable suggestions.**
>
> **Q3**: Can't model like BLIP and along with a model which takes a query and features of the frames from BLIP figure which frames are relevant to the query?
>
> **A3**: Yes, it is possible. **The construction of Task-related Symbolic Memory (TSM) is not limited to a single method**. Our paper only provides one feasible and effective method for constructing TSM. All methods of obtaining task-related information can be integrated into TSM, such as using BLIP to select relevant frames.
>
> **However, some task-related information cannot be stored in the form of a small number of key frames**, such as video object trajectories, segmentations, and other information that requires dense temporal results.
>
> **Q4**: Why TSM is required?
>
> **A4**: Starting from our motivation (in Sec. 1), **firstly, collecting relevant information based on the objectives of the task is a more efficient approach**. Not all information in the video is relevant to the given task. For example, when performing video inpainting, spatial-dominant segmentation information is more crucial than temporal-dominant audio information.
>
> **Secondly, LLMs can be easily distracted by irrelevant and lengthy contexts [b]**. Processing all information in the video at once, as opposed to only processing task-related information, would significantly increase the length of the context and decrease the likelihood of LLMs attending to all pertinent information.
>
> **Thirdly, constructing symbolic memory is more practical for LLMs**, as they can naturally access memory through symbolic language.
>
> [b] Large language models can be easily distracted by irrelevant context. ICML 2023.

---

> > ### Comment · Reviewer_9YSq · 2023-11-23
> >
> > Thank you so much for the detailed clarifications. My concerns have all been allayed.  I'm glad to increase the score to an 8.

---

> > > ### Author Response · Authors · 2023-11-23
> > >
> > > Thank you for your prompt response and your high recognition of our responses! If you still have any questions or further suggestions, please feel free to reply. We welcome any opportunities to improve our paper.
> > >
> > > Once again, we appreciate your effort and thoughtful suggestions throughout the review process!

---

### Author Response · Authors · 2023-11-23
**Looking Forward to Your Feedback**

Dear Reviewers,

We are highly grateful for your time and effort to review our work! We understand that you may have busy schedules, but we greatly value your feedback. Our aim is to gain insights into whether our responses effectively address your concerns and to address any additional questions or points you may have.

We are delighted to see that some of you highly recognize our detailed responses and have raised the rating of our paper. We sincerely appreciate your positive feedback!

We eagerly look forward to the opportunity for further discussion with you. Thank you for your thoughtful consideration.

Best regards,

The Authors

---

### Meta-Review · Area_Chair_sfjA · 2023-12-04

**Metareview:**

Reviewers found this work to be fairly novel and well-written, with good performance relative to comparable LLM-based methods.

In their initial reviews, multiple reviewers raised valid concerns about small size of the evaluation set, only one dataset being tested on, token-efficiency of the method, and overclaiming in the title, etc. with regards to phrase "real-world". The authors performed some additional experiments and analyses, etc., and overall reviewers found their concerns to be somewhat addressed.

However, some weaknesses remain (although not necessarily fatal ones). The paper could have been stronger if various aspects of evaluation were more robust, such as more datasets, more comparisons, etc. The size of the evaluation set is still small (only 18% of NeXT QA val set = 900 / 4996). Even with the addition of TVQA, that's only two out of a number of video QA datasets. In terms of results, this "system paper" puts together quite a number of existing foundation models, plus GPT3.5-Turbo, with the "system novelty" aspects of symbolic memory and MCTS. It outperforms VideoChat by a bit, yet with all its "borrowed power" from specialist sub-models, it underperforms a number of supervised models by a bit as well. It's not a major issue -- but a weakness nonetheless.

Taking the broader view, the long-term significance of the paper is not clear, as it put together a large number of various foundation models in a way that is moderately novel, for results that exceed another LLM-based agent by a bit, and does not exceed supervised SOTA.

Overall, this is a decent paper, and I encourage the authors to continue this line of work, with its system approach, especially the symbolic memory. However, with ICLR being a highly-selective venue, this work is weighed down by some remaining weaknesses and unclear long-term significance.

**Justification For Why Not Higher Score:**

The paper could have been stronger if various aspects of evaluation were more robust, such as more datasets, more comparisons, etc. The long-term significance of the paper is not clear, as it put together a large number of various foundation models in a way that is only moderately novel, for results that only somewhat exceed another LLM-based agent, and does not exceed supervised SOTA.

**Justification For Why Not Lower Score:**

There's fairly unanimous agreement that the work is novel and interesting.

---

### Decision · Program_Chairs · 2024-01-16

Reject